# Music Alters Conscious Distance Monitoring without Changing Pacing and Performance during a Cycling Time Trial

**DOI:** 10.3390/ijerph20053890

**Published:** 2023-02-22

**Authors:** Gustavo C. Vasconcelos, Cayque Brietzke, Paulo E. Franco-Alvarenga, Florentina J. Hettinga, Flávio O. Pires

**Affiliations:** 1Exercise Psychophysiology Research Group, School of Arts, Sciences and Humanities, University of São Paulo, São Paulo 03828-000, Brazil; 2PhD Program in Human Movement and Rehabilitation Sciences, Department of Physical Education, Federal University of São Paulo, São Paulo 11015-020, Brazil; 3Department of Physical Education, Estácio de Sá University, Resende 27515-010, Brazil; 4Department of Sport, Exercise and Rehabilitation, Northumbria University, London E1 7HT, UK; 5PhD Program in Sciences of Rehabilitation, Faculty of Medicine, University of São Paulo, São Paulo 01246-903, Brazil

**Keywords:** time perception, endurance performance, motivation, fatigue

## Abstract

Athletes use their own perception to monitor distance and regulate their pace during exercise, avoiding premature fatigue before the endpoint. On the other hand, they may also listen to music while training and exercising. Given the potential role of music as a distractor, we verified if music influenced the athletes’ ability to monitor the distance covered during a 20-km cycling time trial (TT20km). We hypothesized that music would elongate cyclists’ perceived distance due to reduced attentional focus on exercise-derived signals, which would also change their ratings of perceived exertion (RPE). We also expected that the motivational role of music would also be beneficial in pacing and performance. After familiarization sessions, ten recreational cyclists performed an in-laboratory TT20km while either listening to music or not (control). They reported their RPE, associative thoughts to exercise (ATE), and motivation when they each perceived they had completed 2-km. Power output and heart rate (HR) were continuously recorded. Cyclists elongated their distance perception with music, increasing the distance covered for each perceived 2 km (*p* = 0.003). However, music reduced the error of conscious distance monitoring (*p* = 0.021), pushing the perceived distance towards the actual distance. Music increased the actual distance–RPE relationship (*p* = 0.004) and reduced ATE (*p* < 0.001). However, music affected neither performance assessed as mean power output (*p* = 0.564) and time (*p* = 0.524) nor psychophysiological responses such as HR (*p* = 0.066), RPE (*p* = 0.069), and motivation (*p* = 0.515). Cyclists elongated their distance perception during the TT20km and changed the actual distance–RPE relationship, which is likely due to a music-distractive effect. Although there was a reduced error of conscious distance monitoring, music affected neither pacing nor performance.

## 1. Introduction

Although there is growing availability of sport devices that provide exercise-related information, athletes frequently use their own perception to regulate pace in endurance time trial exercises [1,2]. A successful pacing strategy requires the athlete to self-monitor their elapsed distance or time remaining to the exercise endpoint in order to maximize performance without the occurrence of premature fatigue [1,2]. Despite slight differences in pacing models, it has been suggested that athletes use the progression of their perceived exertion during exercise to estimate the time remaining to the finish line and avoid maximal perceived exertion levels before the exercise endpoint [1,3,4]. Hence, perceived exertion is an important tool for successful pacing regulation, according to these models.

Results from some studies provide support for a perceived exertion-based pacing model, as individuals in these studies reached their maximal or near-maximal ratings of perceived exertion (RPE) at the exercise endpoint, regardless of exercise mode and fatigue progression [5,6]. However, others have challenged the effectiveness of a perceived exertion-based pacing model, as manipulations of sensory inputs changed RPE progression and the athletes’ ability to self-monitor their distance during exercise [7,8]. For example, a light-deprivation study found that cyclists perceived their distance as elongated, reporting a lower RPE when they performed a 20-km cycling time trial (TT20km) in a light-deprived environment [8]. Another important finding was that cyclists also focused less on associative thoughts to exercise (ATE) in light deprivation than in normal lighting conditions, indicating that exercise-derived body sensations were attenuated when light was reduced. Hence, these results evidenced that changes to sensory inputs may influence RPE progression during exercise, challenging the athletes’ ability to self-monitor their distance. Somehow, this effect may be further associated with alterations in exercise-derived body sensations.

Music is a sensory input easily found in real-world exercise scenarios, as athletes from different modalities and performance levels listen to music during exercise bouts in training sessions [9,10,11,12]. For example, different from professional highly-trained athletes who may prefer to avoid distraction from exercise-related bodily sensations [13,14], recreational cyclists may listen to music as an ergogenic aid for indoor and outdoor exercises [14,15,16], as music reduces the focus from exercise-derived aversive sensations and attenuates RPE at comparable exercise intensities [17,18]. Despite controversial results [15], studies have found improved time trial performance with varied effects on pacing when participants listen to music during exercise [19,20,21,22]. Therefore, music is a bodily perception distractor [8,9] with potential motivation effects on pacing and performance [20]. Given its role as a distractor during exercise [11,23], one may hypothesize that music impairs cyclists’ ability to self-monitor their distance due to their reduced focus on exercise-derived bodily sensations and attenuated RPE progression. Somehow, this impaired distance self-monitoring may be associated with music’s motivational effects, boosting pacing and cycling performance [17].

Therefore, in the present study, we investigated if listening to music throughout a cycling time trial affected athletes’ ability to monitor their actual distance. We hypothesized that the distractor effect of music would reduce ATE and lower RPE progression during a cycling time trial, thereby elongating their perceived distance and increasing the error of conscious distance monitoring. However, we expected that the motivational effects of music would improve pacing and performance regardless of impaired self-distance monitoring.

## 2. Materials and Methods

### 2.1. Participants

Ten male recreational cyclists (31.0 ± 4.1 years old, body mass of 69.3 ± 8.1 kg, height of 171.0 ± 4.2 cm, and peak power output of 343.8 ± 20.7 W) classified as P3 level [24] were recruited to participate in this study. They were regularly training and competing in local races by the time the study was conducted. They were non-smokers, free from neuromuscular, cardiopulmonary, cognitive, visual, and hearing disorders, and signed written informed consent forms after clarification about the experimental procedures and risks. This study was previously approved by the Ethics Committee for Research (Protocol 1.605.442) board from the University of São Paulo. All methods and procedures were performed according to the Declaration of Helsinki.

### 2.2. Design

In this crossover study, cyclists were familiarized with its procedures, and they self-selected a music playlist in the first visit before performing a maximal incremental test for characterization. In the second and third visits, cyclists were familiarized with the TT20km, having exercise-related feedback available only in the second visit. In the fourth and fifth visits, cyclists performed the TT20km without exercise-related feedback while listening or not listening to music (counterbalanced order). All sessions were performed at the same time of day (21 °C temperature and 60% relative humidity), and they were interspersed by 3–7 days. They were asked to refrain from stimulant beverages, alcoholic drinks, and intense bouts of exercise for the 48 h preceding the tests.

### 2.3. 20-km Cycling Time Trial

The TT20km was performed on a road bike (Giant^®^, Taichung, Taiwan) coupled with a cycle-simulator (CompuTrainer™ RacerMate^®^ 8000, Seattle, WA, USA), which was calibrated before each test according to the manufacturer’s recommendations. Cyclists performed a standard warm-up (5-min self-paced warm-up and 1-min controlled-pace warm-up at 100 W and 80 rpm) and immediately started the TT20km. Cyclists were free to vary gears throughout the trial. Before trial commencement, they were strongly encouraged to finish the TT20km as fast as possible, but no verbal encouragement was provided during the trial. Although they had no feedback during the trials, they knew the trial would finish at the actual 20 km. Importantly, athletes may shorten or elongate the perception of actual distance, creating a mismatch between actual and perceived distance in different directions (i.e., perception error). Then, a researcher asked cyclists to keep cycling until completing the 20 km if he realized the cyclists were shortening the actual endpoint. Cyclists self-reported RPE and ATE at perceived 2 km partials. Cyclists were informed of their performance times only after the study was completed, thereby avoiding deviations from the intended intervention.

### 2.4. Music Selection

Participants self-selected a 45 min music playlist with songs having a rhythm of approximately 120 bpm, as this music tempo is associated with greater motivation to perform exercise [25]. Trained recreational cyclists may prefer motivational music during exercise, as slow-tempo music seems to be counterintuitive to induce vigor and feelings of self-efficacy [26,27]. Considering that the TT20km would be completed within 40 min, a 45 min music playlist ensured that music was continuously played throughout the exercise bout. The playlist was stored (MP3 device) to be used in the music trial with headphones. The volume was individually adjusted to avoid auditory discomfort. In the non-music session, cyclists wore headphones throughout the trial, but no music was played. To avoid cyclists calculating the time elapsed by counting the songs, they were unaware of the musical order.

### 2.5. Measures and Calculations

Power output was recorded throughout the trial, and values were averaged every 2 km of the actual distance covered for pacing strategy analysis [8]. Performance outcomes were the mean power output and the time to complete 20 km. Furthermore, we calculated the wheel cadence as a measure of actual displacement. Briefly, pedal cadence may not be a reliable measure of displacement in self-paced time trials performed with free gears, as the impact of pedal cadence variations over displacement may be annulled by variations in the gears. In contrast, changes in displacement are directly related to variations in wheel cadence, irrespective of the pedal–gears relationship. Hence, wheel cadence was used as a reliable and objective measure of physical displacement as if the stationary time trial was conducted on an outdoor track. The values measured within the last 15 s of each 2 km of self-perceived distance were averaged and plotted as a function of perceived distance. Due to pacing analysis proposals, power output and wheel cadence were plotted as a function of actual distance.

Heart rate (HR) was continuously recorded throughout the exercise bout with a cardio belt with Bluetooth transmission (Polar^®^, Kempele, Finland). The HR dataset was checked for outliers, defined as those values > 3 standard deviations (SD) from the local mean. Thereafter, data of the last 15 s intervals within each self-perceived 2 km were averaged and plotted as a function of perceived distance. Importantly, some cyclists elongated their perceived distance and reached the end of the trial (actual 20 km) before they perceived 20 km. In these cases, a linear regression model based on HR progression during exercise [8] was used to estimate the values at missing points.

The RPE was measured with a 15-point Borg scale after familiarization with anchors suggested elsewhere [6]. On the other hand, the percentage of ATE was obtained with a 10-point bipolar Likert scale [28]. This scale classified scores from 0 to 4 as dissociative thoughts to exercise and those from 6 to 10 as ATE (5-point scores represented the shift from dissociative to associative). Thoughts related to exercise-derived responses such as HR, muscle discomfort, and breathing were rated as ATE, whereas exercise-unrelated thoughts, such as those associated with daily tasks, personal projects, the environment, etc., were rated as dissociative thoughts. Values close to 0–10% suggest thoughts highly dissociated from exercise, whereas values close to 90–100% suggest thoughts highly related to exercise [29,30]. Cyclists were aware of the distinction between associative and dissociative thoughts to exercise, but they were directed to report only the ATE. The cyclist’s motivation was assessed throughout the trial with a 5-point Likert scale (1–5) that had two antagonistic descriptors of motivation: very unmotivated (1) and very motivated (5) [31]. Importantly, rather than reporting RPE, ATE, and motivation at defined regular intervals, cyclists reported their perceptual responses every 2 km of self-perceived distance. Hence, when they perceived they had completed a 2 km partial, they voluntarily reported these responses. To avoid order bias, a researcher presented the scales in random order. These psychological variables were also plotted against perceived distance. Moreover, we used a linear regression model based on the linear progression of these responses [8] to estimate the values at missing points if cyclists elongated their perceived distance, ending the trial before they perceived 20 km.

Finally, we calculated the error of conscious distance monitoring as a percentage of deviation between perceived and actual distance by plotting actual distance against perceived distance. For example, a cyclist perceiving 2.0 km at 1.90 km of the actual distance would have compressed the actual distance by 5%. We also calculated the distance–RPE index as the ratio between actual distance covered and momentary RPE, thereby indicating how music disrupted RPE.

### 2.6. Statistics

We were unaware of previous studies investigating if music influenced athletes’ RPE-based distance perception. Although studies found large effects on the distance–RPE relationship in running [7] and in cycling time trials [8] using other sensory inputs, we were conservative and assumed a moderate effect of music on this outcome (f = 0.25). Therefore, considering a 2 × 10 repeated-measure crossover design with a minimal power of 0.8, alpha error of 0.05, and moderate within-subjects correlation, 10 participants would be required to detect meaningful effects from music on the overall distance–RPE index (G*Power, version 3.1.9.7, Heinrich Heine University Düsseldorf, Düsseldorf, Germany).

Data distribution was previously confirmed with Shapiro–Wilk’s test. A 2 × 10 repeated-measure mixed model design compared the dependent variables during exercise (power output, wheel cadence, HR, error of conscious distance monitoring, RPE, ATE, and motivation) between music and non-music trials. The covariance matrix that best fit the dataset was used to model the mixed model (Akaike Information Criterion, AIC) criterion), and Bonferroni’s test was used to correct the effects of multiple comparisons on *p*-values. Performance outcomes (time and mean power output) were compared between music and non-music trials through a paired *T*-test. We calculated effect size (ES) according to the statistic test’s family (ƞ2 and Cohen’s d for repeated-measure design and paired-*T* tests, respectively) to confirm the effectiveness of the sample size calculation. However, to allow comparisons with previous studies, we converted them to Cohen’s index (d) and classified them as very small (<0.2), small (≥0.2 and <0.5), moderate (≥0.5 and <0.8), and large effect sizes (≥0.8), as suggested by Cohen (1988) [32]. Analysis was carried out in customized software (SPSS v 21.0, SPSS Inc., Chicago, IL, USA) with the significance set at 5%. Results were reported as mean and SD.

## 3. Results

### 3.1. Perceived Distance and Error of the Conscious Distance Monitoring

No music–distance interaction effect was found in perceived distance (F = 0.24; *p* = 0.10; d = 0.23 small ES). However, cyclists elongated their perceived distance when listening to music, as they covered higher actual distances over each perceived 2 km part in music trials than in non-music trials (music main effect; F = 9.27; *p* = 0.003; d = 1.43 large ES). We also observed that cyclists elongated their perceived distance in both trials (distance main effect; F = 72.31; *p* < 0.001; d = 4.01 large ES).

Regarding errors in conscious distance monitoring, we found no music–distance interaction effect (F = 0.04; *p* = 1.00; d = 0.09 very small ES). However, music reduced the error of conscious distance monitoring (music main effect; F = 5.40; *p* = 0.02; d = 1.10 large ES), resulting in more accurate conscious distance monitoring in music trials (error = 0.42 ± 29.70%) than in non-music trials (error = −6.61 ± 24.65%). In addition, no perceived distance main effect was observed (F = 0.135; *p* = 0.999; d = 0.17 very small ES).

Although no music–distance interaction effects were detected (F = 0.20; *p* = 0.99; d = 0.21 small ES), music disrupted the actual distance–RPE relationship, as cyclists covered higher actual distances to perceive a comparable RPE in music trials than in non-music trials (music main effect; F = 8.40; *p* = 0.004; d = 1.37 large ES). Moreover, there was a progressive increase in the distance–RPE index as the trials progressed (perceived distance main effect; F = 27.66; *p* < 0.001; d = 2.48 large ES). Figure 1A–C depicts the responses related to the error of estimating the distance.

### 3.2. Pacing and Performance

Music did not affect performance, whether it was expressed as mean power output (non-music 204.0 ± 34.2 W vs. music 199.6 ± 42.4 W; *p* = 0.56; d = 0.11 very small ES) or as time (non-music 35.19 ± 2.64 min vs. music 36.07 ± 3.85 min; *p* = 0.46; d = 0.27 small ES). Regarding the pacing strategy, we observed a descending pacing profile in both trials (distance main effect), as power output decreased throughout the TT20Km regardless of whether there was music (F = 5.35; *p* < 0.001; d = 1.09 large ES). Neither music’s main effect (F = 2.14; *p* = 0.15; d = 0.70 moderate ES) nor music–distance interaction’s effects (F = 0.22; *p* = 0.99; d = 0.22 small ES) were observed in power output distribution.

Accordingly, there was a distance main effect (F = 5.09; *p* < 0.001; d = 1.06 large ES) in wheel cadence, showing a reduction in physical displacement as the trials progressed. Neither music’s main effect (F = 0.17; *p* = 0.69; d = 0.39 small ES) nor music–distance interaction’s effect were found (F = 0.50; *p* = 0.87; d = 0.33 small ES). Figure 2 depicts the power output (panel A) and wheel cadence (panel B) responses during the trials.

### 3.3. Psychophysiological Responses to TT20km

Neither music–distance interaction’s effect (F = 0.30; *p* = 0.97; d = 0.26 small ES) nor music as the main effect (F = 3.35; *p* = 0.070; d = 0.86 large ES) influenced RPE progression. However, the main effect of perceived distance showed that RPE increased throughout the trials (F = 29.83; *p* < 0.001; d = 2.57 large ES). Likewise, neither music–distance interaction’s effect (F = 1.16; *p* = 0.32; d = 0.51 moderate ES) nor music as the main effect (F = 0.42; *p* = 0.51; d = 0.31 small ES) influenced motivation, although there was a progressive reduction in motivation as the trials progressed (perceived distance main effect; F = 2.82; *p* = 0.004; d = 0.79 moderate ES). Although no music–distance interaction effect was observed in ATE (F = 0.35; *p* = 0.96; d = 0.28 small ES), ATE reduced progressively during the trials (perceived distance main effect; F = 4.22; *p* < 0.001; d = 0.97 large ES), with lower values in music trials than in non-music trials (music main effect; F = 24.90; *p* < 0.001; d = 2.35 large ES).

Regarding HR responses, neither music–distance interaction’s effect (F = 0.10; *p* = 0.45; d = 0.47 small ES) nor music as the main effect (F = 3.41; *p* = 0.07; d = 0.82 large ES) were observed. However, the main effect of perceived distance showed that HR increased progressively during the trials (F = 2.10; *p* = 0.03; d = 0.68 moderate ES). Figure 3 depicts the psychophysiological responses to music and non-music trials.

## 4. Discussion

As we had hypothesized, cyclists elongated their perceived distance when listening to music. However, instead of a negative effect, music improved conscious distance monitoring, although it was ineffective at changing pacing and improving performance. These results challenge the RPE-based pacing models.

We found that cyclists elongated their perceived distance and increased the actual distance–RPE relationship when they listened to music during the trial, which is likely due to reduced attentional focus on exercise-derived body signals. Indeed, we found that cyclists reported a lowered ATE throughout the trial when listening to music. Interestingly, this effect was beneficial to distance perception, as music pushed the perceived distance towards the actual distance and reduced the mean error of conscious distance monitoring when compared to non-music trials. In this regard, a previous study observed that recreational cyclists failed to accurately monitor (~20%) the distance during a control TT20km, as they elongated the perceived distance [8]. Cyclists of the present study also distorted the conscious distance monitoring in non-music TT20km; however, they did this by compressing it (−6.6%). Together, these results suggest that cyclists naturally fail to accurately perceive the actual distance covered when exercise-related feedback is unavailable, even if sensorial input is unchanged. Somehow, music reduced this error.

The improvement of distance perception with music could be related to the time–displacement relationship of navigation. Considering that time may be symmetrically related to displacement mainly in stationary cycling [33], music tempo may have provided virtual landmarks that allowed cyclists to improve their mental representation of time, thereby allowing them to estimate their virtual displacement (i.e., perceived distance) with accuracy. Importantly, the music tempo did not influence the actual displacement as measured with wheel cadence in the present study. In this sense, given that a variety of internal and external stimuli may play a role as pacemakers [34,35], the cyclists’ internal clock may have taken other stimuli, such as HR, to regulate locomotor rhythmicity. Indeed, the results of HR and wheel displacement were comparable between music and non-music trials, perhaps supporting this hypothesis. The fact that pacing was consistent in both trials, even though conscious distance monitoring was improved with music, could suggest that pacing is regulated through a complex balance between internal and external stimuli rather than solely through the RPE template, as suggested elsewhere [2,4].

A recent study evidenced that music improved performance of recreationally active individuals, as they cycled longer at high intensity when they listened to music [36]. In contrast, we observed that music affected neither pacing nor performance, although it improved distance perception. A recent systematic review that included studies with different exercise modes found that music had a significant but small effect on overall exercise performance [23], although the effects on pacing and time trial performance are controversial. Lim et al. (2009) [22] observed that cyclists showed a faster start pace during a 10-km cycling time trial when music was introduced at the last half of the trial, despite no effect on performance. In contrast, Lima-Silva et al. (2012) [20] found that participants running a 5-km time trial showed a faster start pace when they listened to music during the first 1.5 km of the trial, but performance was improved only if music was introduced during this first part of the run. Perhaps introducing music in specific sections of a trial may be more effective to affect pacing than listening to music throughout the trial, as this may create a motivationally induced behavioral change that takes place within the trial in progress [22]. Indeed, we found that music changed neither pacing nor motivation. The results regarding motivation showed that music triggered no behavioral change, as motivation reduced progressively throughout the trials.

Music’s beneficial effects on pacing strategy and time trial performance may require available task-related feedback [37]. The absence of distance feedback in the present study may have created an uncertainty period that lasted the entire exercise, which would cause cyclists to be less confident about the “right moment” to use this music-triggered psychological booster, as observed elsewhere [38]. The unavailable distance feedback may have prevented the cyclists from speeding up at the end of the trial [8] as usually observed from the ~90% point of a TT20km with available feedback [39]. In contrast to others [20], we left no task-related feedback available as we investigated conscious distance monitoring.

The fact that more accurate conscious distance monitoring with music was ineffective at changing pacing challenges RPE-based pacing models [1,3]. Models based on RPE templates suggest that pacing regulation is a behavioral action that considers the perception of sensorimotor interactions and contextual conditions, and the RPE is a “gestalt” of this process [3,4]. We showed a mismatch between RPE and actual distance as reported elsewhere [7,8]; thus, RPE progressed similarly in both sets of trials, although there were different rates of distance perception with music. One may argue that time and distance are likely regulated through a complex integration of a variety of physiological and environmental pacemakers rather than solely RPE; thus, models considering a more comprehensive integration of physiology and perception may be necessary to explain pacing regulation [3].

### Methodological Considerations

It should be highlighted that distance monitoring in stationary cycle ergometer exercise may be different from distance monitoring in real-world scenarios. We conducted this study with a stationary cycle ergometer because it allowed a more controlled setup. Future studies are required to confirm these results in settings closer to real-world scenarios [40].

In contrast to a previous study [8], we observed that cyclists compressed rather than elongated conscious distance perception during a control TT20km. The fact that this error of conscious distance monitoring may occur in different directions may indicate individual differences in pacemakers [41]. Individual differences may be related to the amount of attention given to time, as reduced attention on time-related issues may induce a loss of retrieved time information and impair the accuracy of time estimation [42]. These results require more investigation with adequate designs.

One may question how these results are applicable to other auditory stimuli commonly encountered in competition and training settings. In humans rather than primates [43,44,45], rhythmicity plays an important role in movement patterns, as rhythm is closely coupled with movement frequency [44,45]. One may argue that other real-world auditory stimuli showing regularity could induce comparable alterations in body movement, thereby changing distance perception. However, most real-world stimuli such as those produced by the audience, athletes, vehicles, machines, etc. are random noise and are unlikely to influence time and distance perception as we observed with music. Music, rather than random noise, plays an important role as a motivational–behavioral trigger of motivation and vigor [46].

## 5. Conclusions

Trained recreational cyclists elongated their perceived distance when they listened to music throughout a TT20km, which is likely due to music-induced distraction. Their elongated distance perception with music altered the actual distance–RPE relationship and improved conscious distance monitoring despite being ineffective at improving pacing and performance.

## Figures and Tables

**Figure 1 ijerph-20-03890-f001:**
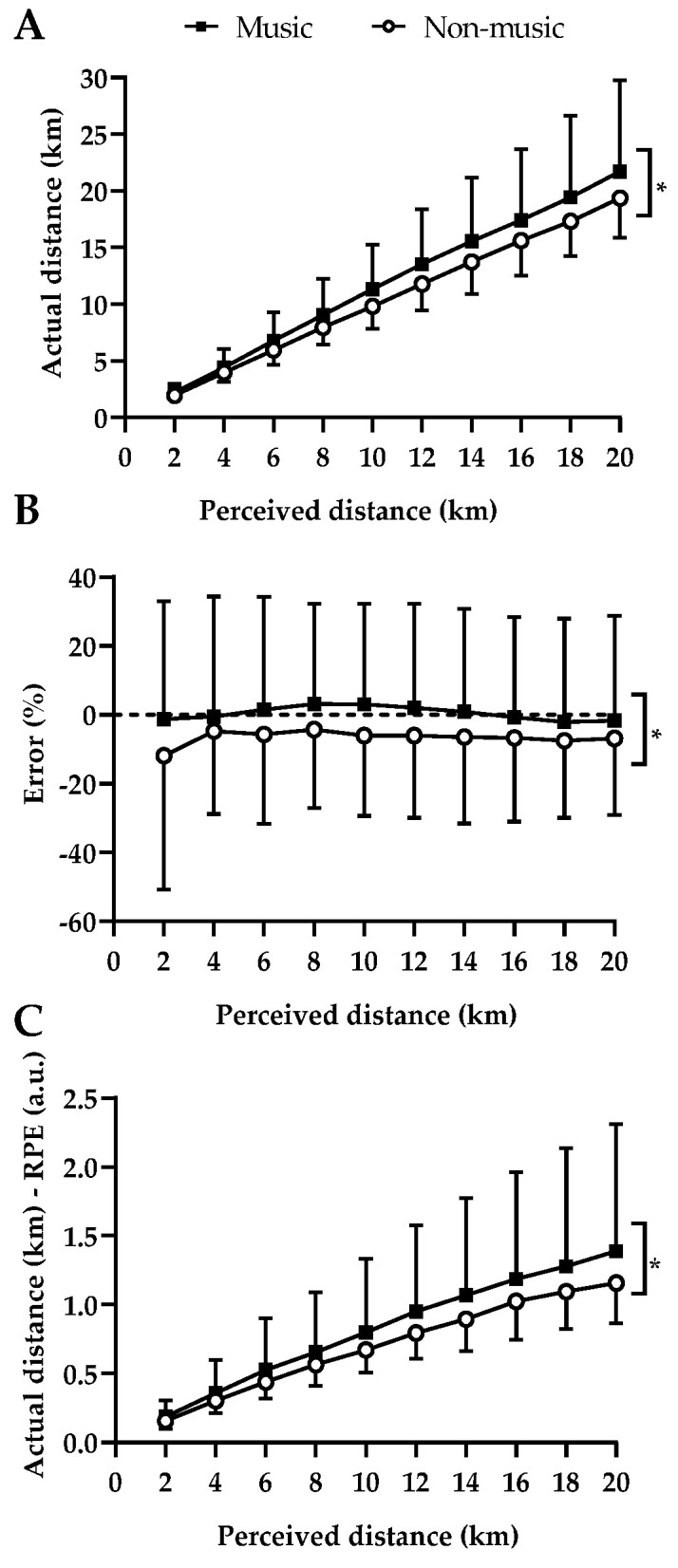
(**A**) Relationship between actual and perceived distance; (**B**) conscious distance error %; (**C**) actual distance–ratings of perceived exertion (RPE) over perceived distance. (*) Main condition effect. A distance main effect was observed in actual distance (*p* = 0.000) and in the actual distance–RPE relationship (*p* = 0.000). Values are mean and ±SD.

**Figure 2 ijerph-20-03890-f002:**
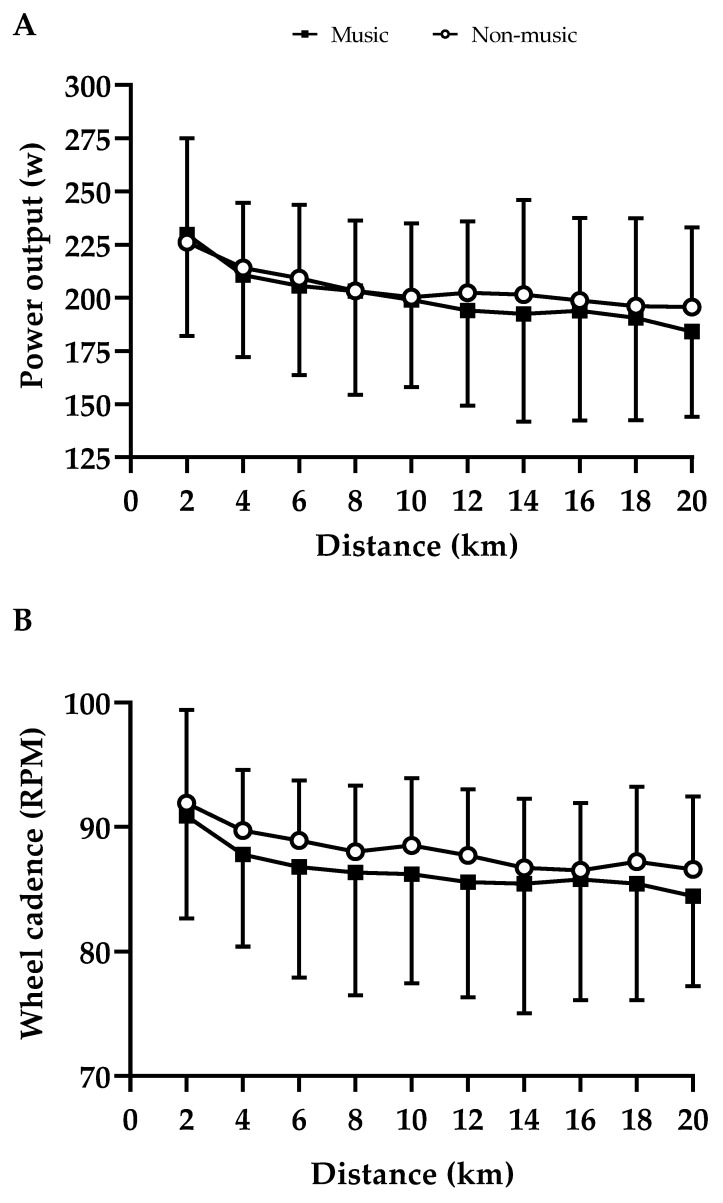
(**A**) Power output and (**B**) cadence during TT20km. A distance main effect was observed in power output (*p* = 0.000) and cadence (*p* = 0.000). Values are mean and ±SD.

**Figure 3 ijerph-20-03890-f003:**
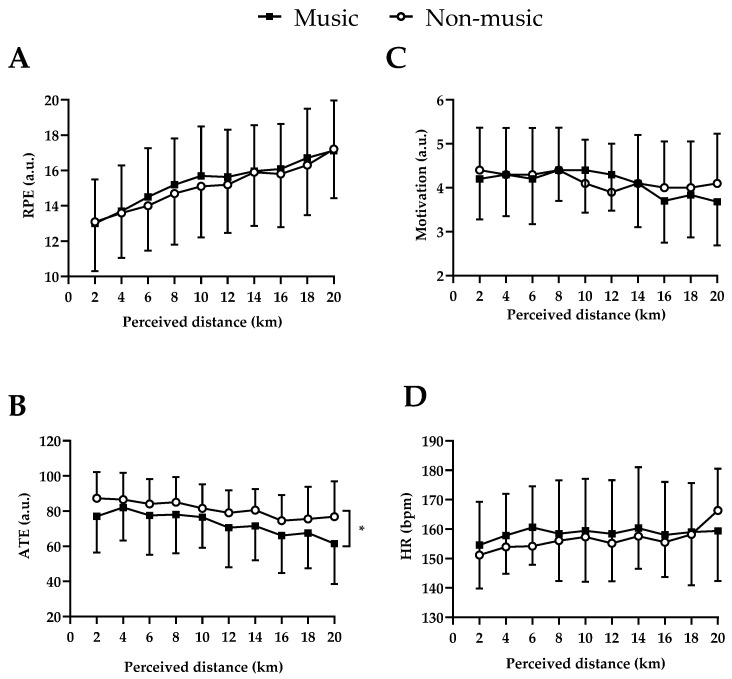
(**A**) Ratings of perceived exertion (RPE); (**B**) associative thoughts to exercise (ATE); (**C**) motivation and (**D**) heart rate (HR) during the TT20km. (*) Condition main effect. A distance-based main effect was found in RPE (*p* = 0.000), motivation (*p* = 0.004), ATE (*p* = 0.000), and HR (*p* = 0.032). Values are mean and ±SD.

## Data Availability

The data presented in this study are available on request from the corresponding author. The data are not publicly available due to institutional policy.

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
