# Peer review of "Music Alters Conscious Distance Monitoring without Changing Pacing and Performance during a Cycling Time Trial"

_ijerph, 2023, doi:10.3390/ijerph20053890_

Round 1

Reviewer 1 Report (New Reviewer)

The manuscript is written well and investigates the effect of music on athletes riding a cycling TT.  The findings are interesting as they, at times, conflict other published research, making them more valuable.  

I would recommend publishing the manuscript with the authors addressing the following minor issues:

1. Should the subjects be described as "recreationally trained" instead of "non-professional"? This is not a big deal, but some may prefer that language.  

2. Do the authors have any data about the athletes' aerobic capacity that could be listed in lines 86-87?  That may help others to understand the level of fitness these athletes possessed. 

3. In line 89, place an "in" between competing and local.

4. In line 126, hyphenate 45-min.

5. Lines 271-278 may need to be expanded. Are there any other studies that found the same relationship between distance-RPE relationship listed in lines 274-276?  This is a quirky finding, but is important to the paper.  The authors cited a paper regarding professionals, which is great. I am a cyclist who uses music often (N.B., I have ridden and competed for over 30 years, but I would have an extremely hard time estimating how long I have ridden in at TT without any feedback. Seeing that music reduced the error is interesting, but I have no idea how it happens.). The authors note this issue in lines 318-324. I have nothing else to add. 

6. The finding regarding RPE listed in lines 326-335 is particularly interesting. I suspect the lack of the use of feedback markers such as time and distance would affect this, too. 

7. Line 374, why would the data not be available publicly? It seems to me that the authors could provide the raw data in a digital file available for downloading.

I recommend that the paper be published with some minor modifications, as noted.  I do not need to see the authors' response to these minor issues, as they are more suggestions rather than demands.  

The authors should be commended for their research by having their manuscript published.  

Author Response

Response to Reviewer 1

Authors: We greatly appreciate all comments and suggestions you have provided.

  1. Should the subjects be described as "recreationally trained" instead of "non-professional"? This is not a big deal, but some may prefer that language.  

Response: Changed, accordingly.

  1. Do the authors have any data about the athletes' aerobic capacity that could be listed in lines 86-87?  That may help others to understand the level of fitness these athletes possessed. 

Response: Unfortunately, we lost all information filled by participants in questionaries and forms when our laboratory was renewed about 1 year after the data sampling. Fortunately, electronic files and ethical consent forms (stored in the office) were unaffected. We agree with you if you say that this is not the best, however we think that the Wpeak obtained in incremental test (together with mean power output and time in the TT20km) may provide a good idea about the participants’ profile. Most important, considering that we aimed to analyze recreational cyclists, the broad range of conditioning status of these cyclists makes us believe that this was not a problem. Please, let us know what you feel.

  1. In line 89, place an "in" between competing and local.

Response: Changed, accordingly.

  1. In line 126, hyphenate 45-min.

Response: Changed, accordingly.

  1. Lines 271-278 may need to be expanded. Are there any other studies that found the same relationship between distance-RPE relationship listed in lines 274-276?  This is a quirky finding, but is important to the paper.  The authors cited a paper regarding professionals, which is great. I am a cyclist who uses music often (N.B., I have ridden and competed for over 30 years, but I would have an extremely hard time estimating how long I have ridden in at TT without any feedback. Seeing that music reduced the error is interesting, but I have no idea how it happens.). The authors note this issue in lines 318-324. I have nothing else to add. 

Response: Thanks!

  1. The finding regarding RPE listed in lines 326-335 is particularly interesting. I suspect the lack of the use of feedback markers such as time and distance would affect this, too. 

Response: We agree!

  1. Line 374, why would the data not be available publicly? It seems to me that the authors could provide the raw data in a digital file available for downloading.

Response: Changed, accordingly.

Reviewer 2 Report (New Reviewer)

Thank you for allowing me to review the article entitled: "Music Alters the Conscious Distance Monitoring without Changing Pacing and Performance during a Cycling Time Trial". In my opinion, the article provides interesting background information relating psychological (ergogenic) aids to sports performance. Also, the article is well written and with all sections well defined. However, I suggest to pay attention to the following points:

In Figure 1, I suggest leaving the panels horizontal and including the X-axis in each panel. Also, in panel A, the X-axis should include the actual distance and the Y-axis the perceived distance. In this way, the standard deviations will match the variables. In panel B, the actual distance should also be included on the X-axis. Panel C is correct.

L226-229: adjust the capital letters A, B and C, and the capital letter "P".

In Figure 2, I also suggest using horizontal panels and including X-axis in both panels.

L245-246: Remove this from the legend: "(*) condition main. A distance main 245 effect was observed in power output (P = 0.000) and cadence (p = 0.000)".

In Figure 3, include X axis in panels A and C.

L265-266: adjust the capital letters A, B, C and D, and the capital letter "P".

L380: Only 25% of the references are from the last 5 years. This means that the introduction, the formulation of the problem and the discussion lack updated scientific support. This is totally improvable. Therefore, I urge the authors to update the references in these sections.

Author Response

We thank you for the opportunity to resubmit our manuscript “Music Alters the Conscious Distance Monitoring without Changing Pacing and Performance during a Cycling Time Trial”. We also thank the reviewers for their time reviewing the manuscript. We appreciate their comments, thus we addressed the suggestions when possible or justified otherwise. We hope we have got the standard required by the journal. All the changes were highlighted in red, and a point-by-point letter is provided below.

Round 2

Reviewer 2 Report (New Reviewer)

I stand by my suggestion for the figures. The X axis should be included in each of them.

Author Response

Response to Reviewer 2

Authors: We appreciate your comments and suggestions; thank you!

  1. I stand by my suggestion for the figures. The X axis should be included in each of them.

Response: Changed, accordingly. Now, the figures in manuscript are presented as below.

This manuscript is a resubmission of an earlier submission. The following is a list of the peer review reports and author responses from that submission.

Round 1

Reviewer 1 Report

Dear Authors,

Find attached the document with my comments. Overall, the paper is well-written and objectives clearly stated. I have some concerns about the definition of some of the important variables (e.g., actual vs. perceived distance) that might becloud the interpretation of the data. These issue can certainly be addressed by the authors in a timely manner.

Author Response

Response to Reviewer 1

Authors: First of all, we would like to thank you for all the comments and suggestions you have done; we appreciate!

PAGE 2

Number: 2 There is no such a thing like recreational athletes. The literature is full of these confusing concepts. An individual might dedicate some time recreationally to training but it does not make him or her an athlete. An athlete is someone who will put training as a priority over everything else to achieve his or her high performance goal. You can use recreational individuals but not athlete.

Response: You are right, the term “recreational” is sometimes unclear and may be confused with those people who use bikes as transport. Instead, we have changed it by “non-professional” cyclists, as they trained regularly and competed in local races by the time the study was conducted. However, they were non-professional ones.

Number: 3 Except if you have read all literature outside English, you cannot affirm that it has not been investigated yet. There are may be 2 or 3 published articles in Mandarin and Cantonese. In addition, the sentence does not bring new pertinente information.

Response: Changed accordingly. We have rephrased this part, let us know what you feel.

Number: 4

Response: Idem.

Number: 5

Response: Thanks for drawing our attention to the reference. The study we should quote is Lim et al. 2009 (32).

Number: 6 See https://peerj.com/articles/6164/

Response: Good study that helped us to support the argument. Thanks!

Number: 7 I would recommend a table with the training profile of the trained cyclists' characteristics. See my comment in the results section on this aspect.

Response: Unfortunately, we cannot attend your suggestion. We lost all information filled by participants in questionaries and forms when our laboratory was renewed about 1 year after the data sampling. We, fortunately, saved the electronic files, but lost the files as papers (ethic consent form was unaffected as this is stored in an institutional file’s room). Thus, what we have is the electronic files and just a remember of their cycling profile shared during talks. Despite we may agree with you that this is not the best, we think that some objective data such as the Wpeak in incremental test and mean power output and time in the TT20km, may provide a good idea about the participants’ profile. Most important, considering the study aimed to analyze non-professional cyclists (earlier defined as “recreational”), the broad range of conditioning status of non-professional cyclists makes us to believe this is not a big problem. Please, let us know what you feel.

Number: 9 According to the paper the authors have cited (20), the participants are classified as trained (class P3 level) not recreational (class P2 level). Please, change the classification throughout the manuscript.

Response: We changed it, accordingly!

Number: 10

Response: Changed, accordingly.

PAGE 3

Number: 1 Trained cyclists

Response: Changed, accordingly.

Number: 4 A polynomial equation could have been used to have a more refined analysis of the pace strategy.

Response: We agree, as we have also used polynomial fitting in other datasets. However, we have seen that polynomial fitting estimates data with different levels of accuracy when the trial is performed without distance feedback (likely due to greater variability in power). Thus, the polynomial fitting has changed the pacing profile in some instances. In this sense, averaging the data within reasonable time windows (such as 2 km) has provided a reliable index of changes in pacing if exercise-related feedback is not available. Thanks!

Number: 6 I would suggest the authors to reword this part since it is the major variable of the study. I am quite confused. I understand that the measurement of displacement is taken at the wheel because variance of pedal cadence does not mean change in displacement in a real word context but on an ergocycle change in RPM means change in distance, except if the authors had allowed the participants to use the shifters.

Response: We rephrased this as suggested. Please, let us know if it is clear now.

Page: 4

Number: 1 "as suggested by"

Response: Thank you!

Page: 5

Number: 1 Participants' characteristics should be reported here. VO2max, training profile (number of training sessions, training volume and intensity). Years of experience and so on.

Response: Please, see our earlier reply. Importantly, we also have no VO2max measures, even though participants performed a preliminary incremental test. That time our old gas analyzer was broken and a new one was brought when the laboratory was renewed. Fortunately, the guideline proposed by de Paw et al. (2013) provides a performance level classification by using performance (WPeak) and physiological (VO2max) indicators.

Page: 8

Number: 1 I would suggest the authors to report some CRF scores of these studies, where possible, to compare with their sample. Response to exercise differs according to fitness. A highly-trained athlete does not behave the same way than a recreational cyclist as the later differs from an unfit individual. so, it would be important to compare the same level of individuals.

Response: This would be insightful. Unfortunately, these studies reported neither VO2max values nor other cardiopulmonary index to be compared with our sample.

Number: 3 Consider reading the following papers on some of these aspects. doi: 10.7717/peerj.6164. eCollection 2019.

Response: thanks for the cue, this gives additional support to some statements.

Page: 9

Number: 1 Add : trained

Response: changed, accordingly

Number: 3 Use participants instead. Subject relates to subjection. In human studies, we do not subject people to participate, we ask for their consent.

Response: changed!

Reviewer 2 Report

Dear Authors, 

Thanks for your submission. It seems that this group has done some work on the area, so I was expecting a better presentation for this work. 

Title points to a design that cannot be supported by the RQ. Abstract seems disconnected as the 2 sentences cannot lead the reader to the hypothesis. There are major leap thoughts that cannot be connected.  Stats and examined variables make no sense, as why to measure all that stuff? I would re-written the abstract from the start.

Intro fails to connect the concepts, it tries, but there are too many concepts to be connected that more paragraphs and info is needed. Main concern is the RQ: How from this

Therefore, in the present study we verified if listening to 76

music throughout a cycling time trial affected the athletes’ ability to monitor the actual 77

distance. 

We to these hypotheses?

We hypothesized that music would lower the perceived exertion and ATE dur- 78

ing exercise, elongating the perceived distance and increasing the error of conscious dis- 79

tance monitoring. We also expected that music may change pacing and performance

And why? if the paper is about theses hypotheses, then rewrite the Intro to address these elements and devote specific paragraphs to connect these all together, so the reader understands the rationale behind the hypotheses and learns what has been done in the area.

Methods need work. I cannot follow the logic behind all these measurements, as I evaluate everything based on the RQ. I see tests that I make not sense to the RQ. The major issue is the Stats and the sample size justification. I used Gpower and got different sample size, and the use of t-tests on 2X2 between-within is not needed. Also I dont see demographics tables and teh quality of the graphs is bad. Also I dont think that SPSS can make such graphs.

Anyway, I stopped reviewing at Results as I have strong objections about the previous sections that would influence the Results and Discussion.

My advice is to make it simple, focus on your RQ, keep your DVs simple, test these, run the appropriate stats for the design and focus on these results and interactions. I am suggesting to resubmit but with major revision - rewriting from the start, not just patches here and there.

Author Response

Response to Reviewer 2 Comments

Thank you very much for the comments, they helped us to improve the manuscript. We appreciate that you have been following us on previous studies, we hope we have got the standard required.

Page 1

Number: 1 to 14

Response: We rewrote the abstract, please let us know if it is clearer now. Thanks!

Number: 14 Intro needs restructure, lacks clarity and flow

Response: We have reorganized the introduction. Please, let us know if it reads well now.

Number: 15 sport

Response: Thanks!

Number: 16 citation

Response: Inserted, accordingly.

Number: 17 are these are done using devices or self-regulatory?

Response: Clarified, accordingly.

Page: 2

Number: 1 so this study is in essence a replication of 6,7

Response: No! None has investigated if music changes the perception-based responses, then changing the individuals’ distance monitoring. This is original.

Number: 2 to 6

Response: We have rewritten the intro, please let us know what you feel now.

Number: 2 not seeing these logic based on the info presented. The info relates to fatigue perception, feelings, not ability to monitor the distance based on RPE. I think this concept needs better writing to connect the concepts, as there is a big loop in logic that I think it fails to do justice

Response: As you can see in the earlier as well in the most recent introduction, we built up the rationale around the distractive effect of music on RPE and, consequently, the self-monitored distance.

Number: 3 what was the explanation of this discrepancy, physiological speaking?

Response: No physiological explanation was provided by those studies.

Number: 6 you cannot verify in this case. You question, examine, whether this....

Response: We have accepted your suggestion and changed the verb. However, we would like to clarify that according to the Merriam-Webster dictionary, “to verify” is used “to attest to the truth or validity of something; “verify implies the establishing of correspondence of actual facts or details with those proposed or guessed at”. The best practice of scientific writing suggests that the verb that directly links manipulation/treatment and outcomes must be weighted according to the type of study (i.e. experimental vs correlational, etc) rather than semantical preference. Considering that experimental studies have the highest scientific power (in scientific terms) to verify the validity of the conclusions (through establishing a cause-effect relationship), verbs that directly link treatment/intervention with the assessed outcomes are fair. Examples: “to verify”, “to examine”, “to analyze”, “to investigate”, etc.

Number: 7 So the main RQ is Music alters distance perception. So I expect to see a design that the same people do a music session and they cover an actual distance you ask them about the perception and one condition that they do not have music and do the same distance effort and ask them their perceptions.

Response: You have made an important point here. In science, we may design different approaches to answer the same research question if we ensure that experimental conditions are controlled and unbiased so that results are reproducible. According to your suggestion (if we understood rightly), in the first session participants would cycle until a predetermined “perceived distance” (“actual distance”?) while listening to music. Then, they would repeat the same distance without music. This design would introduce order-derived bias, thus making the treatment effects to be confounded by the order of testing. In addition, using a freely chosen perceived distance during the first trial would likely reduce the ecological validity (as this is unusual in cycling training). Instead, if you suggested a fixed “perceived” or “actual” distance, then this would be similar to what we have done, however without controlling the order effect through a balanced design.

Number: 8 How do these relate to the RQ? These are different RQ. These hypotheses do not relate to RQ

Response: As you can read from lines 43 to 141, we have hypothesized that music could change the perceived exertion likely due to a distract factor, thus also influencing the self-monitored distance. We further expected that the motivational factor of music would affect pacing and performance. Thus, RPE, ATE, motivation, time and power output were the outcomes.

Number: 9 You dont need that in here

Response: Deleted, accordingly.

Number: 10 why such a test? relevance with the RQ?

Response: This test was used for characterization proposals.

Number: 11 2 visits for familiarization? Do they need even one as they are already cyclists P3 level?

Response: Yes, you are right; studies have shown that 1 familiarization may be enough to get trained cyclists acquainted with cycling time trials. However, we decided to have even higher rigor to control for experimental error-derived bias, thus we performed 2 familiarizations with the time trial (with and without exercise-related feedback).

Page: 3

Number: 1 why?

Number: 2 why? how is this related to RQ?

Response: Any available exercise-related feedback may allow cyclists to improve their accuracy in counting the perceived distance; this would increase the error in our design.

Number: 3 what about tempo? did you select the music, was the music the same? Since the music is the intervention, it needs to be controlled and it seems that are not - let it be free ?

Response: Please, read the “2.4 Music selection” section.

Number: 4 so no controlled pace-effort?

Response: No, this was a time trial.

Number: 5 did they have an access to the monitor? if they are experienced due to fatigue - pace that they follow then they have a good estimate about the covered distacne

Response: No. As mentioned in the “Design” section, they had no exercise-related feedback.

Number: 6 lost you - not in here, maybe stats? why do you need this?

Response: You are right, it was unclear. Briefly, we plotted the psychophysiological responses as a function of the perceived distance (instead of actual distance) to analyze the data. However, given that cyclists may compress the perceived distance, they could reach the actual 20 km before the perceived 20 km. When it was the case, we used a linear regression (based on the progression pattern of these variables during a 20 km time trial) to estimate the values at missing points. We have clarified this and replaced this information on “2.5. measures and calculation” section. 

Number: 7 why?

Response: This strategy allowed calculating the individuals’ distance perception, as they had to self-report their RPE at every 2 km.

Number: 9 okay, but i was wondering if this could underestimate or overestimate the rhythm of the cyclists as their actual strokes may be faster or slower and this lead to differences in their performance. The best approach was to get a song that matches their respective strokes

Response: Good point. Your strategy would have provided a nice interindividual control. However, this may not have been the best control for those cyclists preferring low-beat songs (< 100 bpm), as motivational songs have cadences rather above 100 bpm (~120 bpm). Given that one hypothesis was that music-triggered motivation may influence behavioral-based exercise decisions, we decided to control for this confounding factor. Although agreeing with you that some cyclists may have preferred to listening a bit slower song, our methodological approach allowed controlling for the within-between subjects variability.

Number: 10 not in here, here tell us how you did it not why. use intro for that

Response: This is a methodological justification.

Number: 11 bu this is a sensory distraction -

Response: Sorry, we did not get what you mean.

Number: 12 relevance to the RQ?

Response: This is one of the outcomes directly related to the hypothesis (see reply to inquiry 9). We agree that this sentence was inaccurate, thus we removed it.

Number: 13 Therefore, in the present study we verified if listening to music throughout a cycling time trial affected the athletes’ ability to monitor the actual distance. How does this relate to RQ? The hypothesis need to relate to RQ

Response: As highlighted in the original version, “We hypothesized that music would lower the perceived exertion and ATE during exercise, elongating the perceived distance and increasing the error of conscious distance monitoring”.

Number: 14 why?

Response: Power output was recorded throughout the trial and values were averaged every 2 km of the actual distance for pacing strategy analysis.

Number: 15 why?

Response: We calculated the wheel cadence as a measure of actual displacement. Thus, despite being a stationary cycle-simulator, this index indicated the actual displacement if the trial was conducted on an outdoor track. Pedal cadence may not be a reliable measure of displacement in self-paced time trials performed with free gears, as the impact of variations in pedal cadence over displacement may be annulled by variations in the gears. In contrast, changes in displacement are directly related to variations in wheel cadence, irrespective of the pedal-gears relationship.

Page: 4

Number: 1 dont follow....10 cm scale?

Response: Corrected (it is a 10-point scale). Thanks.

Number: 2 why all these?

Response: Presenting the scales within the same order may induce bias, as participants may conditionate RPE responses according to ATE responses (and vice versa).

Number: 3 this is the only measurement that relates to RQ

Response: We disagree. As highlighted in the introduction, “We have hypothesized that music would lower the perceived exertion and ATE during exercise, elongating the perceived distance and increasing the error of conscious distance monitoring. Thus, RPE (perceived exertion), ATE, perceived distance and error of distance monitoring are all outcomes related to the research question.

Number: 4 not related to RQ

Response: We disagree. According to different theoretical models, athletes use their perceived exertion (RPE) to self-monitor the distance that has been covered. Thus, the relationship between RPE and distance is the primary outcome to be considered. Hence, we estimated the sample size through the distance-RPE index effect size.

Number: 5 This is not your RQ

Response: Sorry, we did not get your point here. We hypothesized that the distractor effect of music would lower the perceived exertion during exercise, elongating the perceived distance and increasing the error of conscious distance monitoring. Therefore, the distance-RPE relationship was the primary outcome used to calculate the sample size (distance-RPE index).

Number: 6 what is the 2 by 10? 2 groups 10 measurements?

Response: Yes, it was a 2x10 repeated-measures ANOVA (light-deprivation vs lighting by 2km, 4km, 6km… 20km distance parts)

Number: 7 This should be between-within design such a design with same values, 2 groups 1- measurements, corr 0.5, non sphericity 1, gives me 14

Response: Assuming that correlations between trials in crossover designs are naturally stronger than parallel group designs, we used the upper borderline of moderate correlation range (0.70).

r < 0.3 None or very weak

0.3 < r <0.5 Weak

0.5 < r < 0.7 Moderate

r > 0.7 Strong

Moore, D. S., Notz, W. I, & Flinger, M. A. (2013). The basic practice of statistics (6th

ed.). New York, NY: W. H. Freeman and Company

Number: 8 these are 8 not 10

Response: These were the outcomes (not sure if you considered them as the factor levels).

Number: 9 hmmm, why? this a 2x2, 2 trials X 2 intervention between-within

Response: Instead of measures that repeated over time, mean power output and time had only one discrete value in both conditions. A paired student T-test is adequate for this proposal.

Number 10: p <0.05

Response: Yes, p < 0.05 means < 5%

Page: 5

Number: 1 I quit reviewing in here as I have objections on how Intro and Methods were presented, and most importantly with the sample size and stats used.

Response: Ok. We used your comments to improve our communication for the reader, the study is quite challenging as it involves perceptions of the distance displaced assessed through the participants’ self-reported perceived exertion.

Number 2: demographiscs?

Response: We did not get what you mean.

Number: 3 quality is bad - are these from SPSS?

Response: We have uploaded figures as recommended by the journals’ norms. We think the figures will have enough quality in the published article.

Round 2

Reviewer 2 Report

Dear Authors, 

Thanks for addressing my comments and revising the manuscript. Sadly in the current state I dont feel confident moving forward. The main issue that I addressed in the previous submission is the clarity on presenting the concepts, design, methods, and stats. You have way too many RQ to be addressed with one design and presented in one paper. I suggested to split it and make 2-3 papers so there is flow and the reader can follow what and how you did things and comprehend the Discussion.  

It seems to me like a Thesis or PhD project that with one research design many things are attempted, but when it comes to publication, it is not possible to present all of them due to word limitations.  

Author Response

Response: Thanks for your time reviewing the paper, we respect your thoughts. However, we totally disagree with some of them, as they may threaten the scientific integrity of the study. The introduction is quite clear in setting objectives and hypotheses. As you can see, we set what is known during the two first paragraphs (describing the pacing model and its controversies), then we set what is still unknown during the last two paragraphs (ending with question and hypothesis). We made clear what our manipulation is (“listening to music during a cycling time trial”) and what the outcomes are (“distance self-monitoring”, “perceived exertion”, “focus” and “motivation”). Different from a Thesis in which parallel studies come together to build up an intellectual proposition around a theme of a field, the present paper presents a clear and limited question and hypothesis (Could music change the cyclists’ ability to self-monitor the distance in a time trial due to a distractor effect on the perceived exertion?). Furthermore, the number of outcomes is determined by the strength of the rationale-driven hypothesis and the operational availability in doing so; it is not a mere word limitation question. Therefore, the suggestion to split the data into 2-3 papers is concerning, as the practice of “salami science” may threaten the scientific integrity. Finally, the study was cautiously designed to avoid bias from different sources; then we used a randomized balanced order to avoid order effects, familiarized participants with procedures to avoid learning effects, reported all outcomes to avoid selection of the reported result, etc. Some of your initial suggestions would have put biases in the design.